# Prophage in Phage Manufacturing: Is the Risk Overrated Compared to Other Therapies or Food?

**DOI:** 10.3390/antibiotics9080435

**Published:** 2020-07-22

**Authors:** Gabard Jérôme

**Affiliations:** BioMaje SAS, 5, Chemin du Vercors, 49124 Sant Barthélemy d’Anjou, France; jerome.gabard@biomaje.fr; Tel.: +33-06-07-24-85-19

**Keywords:** bacteriophage, prophage, manufacturing, risk, antibiotics, fecal matter transfer (FMT), vaccine, probiotic, food, biologic.

## Abstract

The rehabilitation of lytic bacteriophages, as living and replicative biological therapeutic agents, is only 2 decades old in western countries, compared to other therapeutic approaches using chemicals and inactivated or alive biologicals. This paper attempts to provide arguments to address prophage content issues in phage pharmaceutical preparations from a regulatory perspective. The author rebalances the risk associated with the presence of prophages in their pharmaceutical preparations in comparison (i) to lysogenic phages and prophages contained in various therapeutic anti-infective treatments, as well as in food or probiotics, (ii) to adventitious whole retroviruses or fragments contained in vaccines, and (iii) to the massive release of lysogenic phages and prophages induced by antibiotics usage.

## 1. Introduction

Phage therapy started [1,2] in 2017 within the western modern pharmaceutical industry. As the 20th century saw pharmaceutical chemistry rise, the concept of treating with a single, pure, and well-characterized chemical active ingredient became the trend [3]. Bacteriophage suffered from production quality issues [4], and the controversies about their true reality placed them in the category of strange, unpredictable, and inexplicable treatment, until electronic microscopy revealed their true biological nature [5] in 1940. Meanwhile, as antibiotics were discovered from biological organisms and chemically and properly characterized drugs, at reproducible concentration, standardized purity levels and contaminant traces started to rule the world of curative treatments [6].

The emergence of inactive biological drugs started to change the trend in the 1980s through the development of complex, but inert, molecules (recombinant proteins, therapeutic monoclonal antibodies, peptides, DNA, etc., often synthesized by microorganisms [7]. Although the active ingredient is a fixed molecule, the production tool is often a living factory more prone to variability than chemical synthesis. Regulatory agencies had to adapt guidelines and rules to these new treatments [8].

A third revolution occurred in the late 1990s with the arrival of curative therapies, using alive organisms on both production and treatment sides, as described in Table 1.

Biological variability now encompasses both production material and therapeutic treatment. The regulation adapted to fixed, chemical, or biological antibiotic molecules cannot apply anymore. Product variability from batch to batch is almost inevitable [9].

As part of it, the presence of prophages in a manufacturing strain, which accidently contaminates a bacteriophage drug batch, is identified as an issue for phage therapy [4]. Regulatory agencies are hampering the approval of phage therapy by insisting on the absence of prophages from phage preparations used to treat patients. This insistence sets a double standard, because many other approved therapies allows for far greater levels of viral release or contamination without greater evidence of safety.

## 2. Opinion

**Prophages in bacteriophage manufacturing**: Phage therapy uses lytic phages, which multiply cleanly in their production strain [4]. In other words, they do not leave any DNA from their own genome or embark DNA from their host during their replication cycle. 

Contrary to lytic phages, temperate phages can integrate into the chromosome of a bacterial host upon infection, where they can reside as quiescent genetic material [10]. When the strain is stressed, such prophages may start replication. During that cycle, they may embark DNA from their host and transfer it to other bacterial strains. These host DNA fragments may carry virulence, toxins, antibiotic resistance, stress tolerance, and novel metabolic pathway genes, which provide significant adaptation advantages to their hosts [10]. In a way, prophages infecting prokaryote bacteria are comparable to retroviruses infecting eukaryote cells: each of them target a specific cell organism, remain dormant in their hosts, and are expressed when the host is stressed by various chemical, physical, or biological factors. For instance, when bacteria are stressed by Ciprofloxacin or Trimethoprim antibiotics [11], prophages are released, in a similar manner to herpes simplex virus type-2, which is via UV radiation [12]. 

Comparative genomics demonstrated that bacteria and bacteriophages are coevolving. This process is most evident for pathogenic bacteria, where the vast majority contain lysogenic prophages in their DNA. Therefore, the bacterial strains that are used to produce bacteriophages are likely to contain prophages [11]. High yield production in fermenters is a stress for the phage manufacturing strain; therefore, the process may lead to significant prophage release into the culture media [4]. Consequently, prophages may then be found in a phage therapy suspension batch used to treat patients.

Bacteriophages used to combat antibiotic resistance were classified as drugs in 2013 [13]; therefore, their manufacturing procedures must comply with pharmaceutical good manufacturing practices (GMPs), including stringent quality controls (QCs), such as detecting and eliminating the risk of prophage contamination introduced by bacteria master cell lines [14].

**Vaccines containing retrovirus contaminants**: One must notice that for prophylactic medicine, the adoption of biologicals, although inactivated, took place much earlier than that in the alive biological drug world. [15]

Inspired by Edward Jenner in the late eighteenth century and his protection against smallpox through cow vaccinia benign disease, Pasteur stated in 1881 the principle of vaccination: a "weakened viruses with the character of never killing, of giving a benign disease, which preserves from fatal disease". The list of vaccines against pathogenic infections with inactivated or killed microbes started with rabies at the end of the nineteenth century, then diphtheria, tetanus, typhoid, and tuberculosis in the 1920s. Vaccines against epidemic viruses, yellow fever, influenza, and polio rose between the 1930s and 1960s. Since then, with notable thanks to biotechnology, meningitis, hepatitis B, and the papillomavirus were amongst the latest targets [15].

This repeated successes in making efficient and reproductible prophylactic treatments against most major human (and animal) infections made of biological inactivated germs a fully integrated tool of our pharmacopeias.

Nevertheless, although vaccines do not generally bear adventitious contaminants, contamination of cell substrates that are used to produce them have been reported [16]. For instance, in 2010, independent researchers detected DNA fragments of porcinecircovirus (PCV) in Rotarix®, manufactured by GlaxoSmithKline, and in RotaTeq® vaccines produced by Merck. 

After thorough investigation and identification of trypsin as the likely source of contamination, the European Medicinal Agency (EMA) Committee for Medicinal Products for Human Use considered that the PCV findings did not present a threat to public health and, consequently, that there was no need to restrict the use of the vaccines [16].

**Viral risk monitoring is difficult in living human transplants**: Alive biological transplants using tissue graft skin [17], organ transplants, or the newest cell therapy [18] refer to the transfer of body parts or cells from a compatible human donor toward a compatible human recipient.

In homograft, the human material from a patient is used to repair an injury, a failure, a defect, or an anomaly of another injury. Material transfer remains mostly intraspecific, i.e., within human genus, even though non-permanent animal material is sometimes temporarily used, such as pig origin skin graft for severe burn patients. In autograft, the biological material comes from the patient’s own body; therefore, the contamination risk is mostly nosocomial.

Guidelines have been published to mitigate the risk of virus transmission, especially retroviruses through homograft organ transplantation. For the EMA, baseline serology for risk factors for transplantation outcome should include cytomegalovirus (CMV), the hepatitis B virus, hepatitis C virus, Epstein–Barr virus, human immunodeficiency virus, polyoma virus, and Herpes simplex virus [19]. The FDA (Food and Drug Administration) for the USA follows identical recommendations [20]. 

Hence, serology, microbiology, and detailed questionnaires targeting disease-patient history support the choice of the donor. Sometimes, molecular techniques are added to evaluate the presence of retroviruses in the transplants, although such technical approaches are not generalized. Even so, the molecular tools used are not powerful enough to alleviate the risk of recipient contamination. The major hurdles include the worldwide epidemiology shift of infections, for instance with COVID-19, increasing antimicrobial resistance, suboptimal assays for microbiology screening of organ donors, and difficulties in detecting viruses [21].

Nevertheless, regulatory authorities consider that the benefit of organ transplantation for an expecting recipient makes the risk of viral contamination, introduced by the transplant, bearable and manageable using long lasting antiviral treatments, although patients are severely immunosuppressed to avoid graft rejection [22].

**The black hole of fecal matter transfer:** When it comes to interspecific alive biological medicines, the panoply of non-human living organisms used for treating human pathologies is still narrower. A fecal microbiota transplant (FMT) would be classified as hazardous by the fundamentalist preachers of zero safety risk, according to our current level of analytical tools and possibility of risk evaluation.

The millennium old Chinese therapeutic arsenal (the yellow soup) FMT was only tested in 1958 by Western Medicine, leading to the first randomized controlled and successful trial against *Clostridium difficile* in 2013 [23]. With FMT, tens of thousands of unknown bacterial species are transferred from a qualified donor to an ill patient, including yeast, fungi, viruses, and trillions of bacteriophages. Phages may play a major role in *C. difficile* (CDI) patients. Indeed, CDI patient enteric virome dysbiosis seems characterized by an increased *Caudovirales* phage abundance, opposing their decreased diversity, richness, and evenness [24]. In addition to *C. difficile* eradication, randomized controlled studies have shown FMT to be somewhat effective in treating ulcerative colitis, irritable bowel syndrome, and hepatic encephalopathy and have shown FMT to be beneficial for the eradication of multidrug-resistant organisms and graft-versus-host diseases [25].

Although putative donors are qualified through health, social, and travel questionnaires, clinical assessments, and microbiological testing [26], the quality of feces for FMT is far from being a 100% controlled process. Even current metagenomic tools, which may be used to qualify the microbial population of a donor, are ineffective for identifying the molecular details of each stool inhabitant [24]. They certainly skip plasmids, viruses, or prophages carrying antibiotic resistant genes, lysogenic phages, or poorly represented unknown pathogenic bacteria, leaving the door open to a transfer of them and/or their genetic material to the donor. 

The recent bacteremia of two patients following FMT and leading to the death of one of the patients [27] by an ESBL (Enlarged Spectrum Betalactamase) *Escherichia coli* is the proof that a bullet-proof screening process is not implemented yet. However, clinical trials continue, as FMT seems full of promises.

**Where probiotics and pharmabiotics join the scenario:** Probiotics and prebiotics [28] have been used for decades to support human health and the food industry. Most current probiotic bacterial species are lactic strains, *Bifidobacteria*, or, to a lesser extent, *Bacillus* or *Streptococcus thermophilus*. 

Dairy products are a reservoir of antibiotic resistance [29]. Following standard fermentation processes via lactic ferments, genes conferring resistance to tetracycline, erythromycin, and vancomycin have been detected and characterized in *Lactococcus lactis*, *Enterococci*, and *Lactobacillus* species isolated from fermented meat and milk products. Commensal bacteria, including lactic acid bacteria (LAB), may act as reservoirs of antibiotic resistance genes, such as those found in human pathogens [29]. Millions of lysogenic bacteriophages are released by lactic bacteria during fermentation, accumulate into a tank [30], and become a source of antibiotic resistance gene transfer. In addition, prophages have been shown to be released from *Lactococcus lactis* by antimicrobial peptide application [31].

In the case of diet pills and other food complements, such as cheese, yogurt (including fermented foods, such as aban, lassi, kefir, skyr, and other buttermilk), plain milk, and other fermented foods, no authority imposes monitoring of the presence or absence of intact or fragmented phage or prophage material in these products. Whether pasteurized or raw, taken altogether, a great diversity of phages is naturally present in the raw milk ecosystem, thus the absence of phages in dairies is unreachable [30]. An averaged human being eats billions of phages in their daily dairy supply [32].

Pharmabiotics, a new class of bacteria with medicinal properties, have been developed in the last 15 years. Compared to bacteria probiotics that can barely advertise a health claim, pharmabiotics are true drugs with medicinal revendications. For instance, RBX2660 (the first mix of beneficial *Clostridium* bacteria targeting microbiota) has been granted approval by the FDA for clinical testing in *C. difficile* intestinal infection [33]. The product has been classified as a drug instead of a diet probiotic. 

One can assume that the manufacturing of pharmabiotics will be similar to those of probiotics, even though pharmaceutical good manufacturing practices (GMPs) will have to be truly applied. After more than 70 years of manufacturing optimization, probiotics production processes and quality controls are aspiring toward pharmabiotics production. Today, there is no request to check the presence of prophages in the millions of probiotics pills sold daily in the world. But what is the position of health regulatory authorities toward the latest pharmabiotics?

**Antibiotics are potent inducers of prophage release:** Antibiotic resistance emergence risk needs to be documented when filing for market authorization of a new antibiotic. The 2019 European Medicine Agency (EMA) guideline “on the evaluation of medicinal products indicated for treatment of bacterial infections, Rev.3” says that, “for test antibacterial agents of a new class, in vitro susceptibility studies should assess the potential for cross-resistance to occur between the test agents and licensed agents of other classes” [34]. In addition, clinical trials of antibiotic drug candidates must address their impact on the control of major antibiotic resistance forms [35].

Various studies have shown that antibiotics induce or boost lytic phage and prophage release upon treatments. A non-exhaustive list of publications is given below: For *Staphylococcus aureus* under strong antibiotic selection [36];For multidrug-resistant *Stenotrophomonas maltophilia* in response to ciprofloxacin stress [37];For *Clostridium difficile* under pressure of profloxacin, moxifloxacin, levofloxacin, or mitomycin C [38];For enterohemorrhagic *Escherichia coli* O157:H7 by [39];For four isolates of fibrosis epidemic strain of *Pseudomonas aeruginosa* in cystic fibrosis induced by norfloxacin, tobramycin, colistin, ceftazidime, meropenem, or ciprofloxacin [40] at various levels depending on the antibiotic.

When it comes to antibiotic small molecules or peptides, no data regarding their potential effect on the release of lysogenic bacteriophages or prophages from patient bacterial microbiotas in reaction to treatment is requested by regulatory agencies toward filing for market authorization [41].

## 3. Discussion

In bacteriophage manufacturing for human therapy, the growing regulatory trend is to recommend that strains used to make master and working bacterial lines be exempt of lysogenic bacteriophages or of prophages. The reason is that they may carry antibiotic, virulence, and/or toxin genes and can be horizontally transferred to other the bacterial strains of a patient. The French Regulatory Agency (ANSM) tries to elaborate a pragmatic approach to rank prophage risk in five classes, as described in Table 2.

In addition, when the bacterial strain comes straight from a patient (as in the case of salvage therapy), although it should be characterized as much as possible (due to the presence of toxins and virulence genes, etc.), ANSM is more flexible when using it because it already expresses toxins or virulence factors in the sick patient. 

Nevertheless, the comparison with other medicinal and health application fields makes it difficult today to display such different risk evaluation criteria when it comes to vaccines, FMT, probiotics, antibiotics, or fermented food. 

Thousands of years of consumption of cheese or yogurts containing many kinds of pro/lysogenic phages without unusual or visible secondary effects on consumers is a green light for leaving things as they are. Logically, new probiotics and diet supplements containing lactic ferment and phages are not considered a danger for consumers. This is fair because their benefit toward alimentation is far greater than exposing human populations to the risk of eating prophage DNA.

Since the earliest days of vaccine manufacture, there have been cases where laboratory studies provided evidence for the presence of adventitious agents in a marketed product. For instance: (a) SV40 in polio vaccines; (b) bacteriophage in measles and polio vaccines; (c) reverse transcriptase in measles and mumps vaccines; and (d) entire PCV or DNA sequences in rotavirus vaccines [39]. As science and detection means progress, they may be eliminated [16]. However, in agreement with years of practice and a lack of related safety incidents, some of them, such as the PCV porcine virus, are left inside vaccines because they are considered harmless to humans [42].

Antibiotics, which catalyzes pro and lysogenic phage release [43], gene shuffling between bacterial species, and antibiotic resistance genes horizontal transfer have been in use for over 70 years. During the first decades of use, technologies were not available to detect their effects on microbiota flora. Today, we have ways to measure their impact on prophage release, but regulatory health agencies do not ask for this, because, overall, albeit antibiotic resistance emergence, years of practice have demonstrated their efficacy.

Using FMT for gut pathological indication may be considered less risky than injecting a bacteriophage into the blood stream. This may apply when considering a patient with a healthy and fully operational gastrointestinal tract. However, recipients are far from belonging to this category. Their intestinal barrier is quite deteriorated and permeable to many microorganisms, including bacteriophages, prophages, viruses, bacteria, etc. Microorganisms can cross the intestinal barrier and invade other body tissues or the vascular system, leading to severe pathologies [44]. Should we stop FMT from being used? Certainly not, because clinical studies demonstrate their benefits for curing *C. difficile* infected patients and other inflammatory bowel diseases.

## 4. Conclusions

Overall, the ratio between the benefit of phage therapy compared to the risk associated with batch content prophage contamination should be the driver. Do regulators influenced by meticulous research experts raise the bar too high for GMP phage manufacturing to succeed? Are we losing the big picture? After all, phage therapy is 100 years old. Don’t we have some historical experience prior to imposing manufacturing and safety constraints that are not applied to other therapeutic areas? 

## Figures and Tables

**Table 1 antibiotics-09-00435-t001:** Therapies making living treatments from living organisms.

Biological Therapy Type	Production	Treatment
Cell Therapy	Human body	Modified human cells
Fecal Matter Transfer (FMT)	Native Feces from donor	Screened feces to recipient
Bacteria therapy	Microbiota	Intestinal Microbes
Phage therapy	bacterial strain	Bacteriophage

**Table 2 antibiotics-09-00435-t002:** Prophage management risk ranking, according to the French Regulatory Agency (ANSM).

	Case Type	Risk Level
1	Ideal	Prophage-free bacterial strain
2	Intermediate	Presence of prophage(s) carrying gene(s) that do not have code(s) for character(s), having an impact on the efficacy and/or safety of the treatment
3	Degraded	Contains a prophage carrying an impact on gene(s), but which is (are) not (or barely) expressed
4	Insufficient	Contains a prophage carrying an impact gene that is expressed
5	Unacceptable	Contains a non-eliminable lysogenic phage

The cases ranked from 1 to 3 (above the bold black line) are acceptable. The cases below are not.

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
