# Peer review of "Prophage in Phage Manufacturing: Is the Risk Overrated Compared to Other Therapies or Food?"

_antibiotics, 2020, doi:10.3390/antibiotics9080435_

Round 1
Reviewer 1 Report
This is a difficult paper to evaluate. The point of the paper is that phage therapy is being held to higher standards than many other approved treatments; this is a fair and useful point. However, the audience for the paper isn't clear -- it might be only the agencies that regulate phage therapy (although advocates of phage therapy may want to use the points here in applying pressure to the regulatory agencies).
The writing of the paper, especially its organization, is poor -- the organization is effectively backwards. The paper starts out with a review of phage therapy, territory that has been covered by so many other reviews that it will dissuade many reviewers before they get to anything novel. Then, the paper launches into a litany of approved (non-phage) treatments without informing the reader of why it is doing so. At the very end, it gets to the main point -- that prophage contamination of phage preparations should be relegated to a minor issue because there are far worse problems facing other kinds of approved treatments.
Overall, the paper could be streamlined to be 1-2 pages long (of text), skipping the historical review, and getting to the main point immediately. The list of approved treatments that seem more 'egregious' than prophage contamination of phage preps can be relegated to a long, detailed table, with the text providing selected highlights.
Regulatory agencies have been 'hard' on phage therapy for more than just prophage contamination. The points in this paper likely apply to other regulatory hurdles that have been applied.
Jim Bull
Author Response
Dear Jim
Thanks a lot for reviewing the draft of my paper on prophages in phage manufacturing.
I shorten the introduction and made it more focused on the issue that I want to address in the paper: quality constraints on prophage content in phage drug manufacturing.
Comparisons with other approved treatments has been maintained after informing the reader why.
I also toned down some paragraphs upon request from another reviewer and corrected English mistakes. Reference to papers has increased to illustrate better some of my statements.
Overall, I made quite a bit of changes, including an extra table to get my point clearer.
It is true that the paper is for regulatory agencies and for helping any public or private organization that want to get approval of a manufactured phage batch for clinical application. I whish that when I was submitting files for clinical approval of phage therapy trials that such papers would have been available in the literature to support our regulatory dossiers.
I appreciate the time you spent to review the draft of this paper and want again to warmly thank you.
Best regards,
Jérôme
Reviewer 2 Report
The manuscript “Prophage in phage manufacturing: is the risk overrated compared to other therapies or even food?” by Jérôme Gabard is an interesting view on the current discrepancies for risk evaluation of phage therapy compared to other therapies such as FMT and probiotics. I believe this opinion has timeliness and it addresses a very important topic that deserves proper discussion. I would recommend its publication after the few comments below are addressed.
Comments:
- While this is an opinion paper, and a quite well written one, I think it would benefit from some toning down of the opinionative view.
- Lines 59-63, I appreciate the comparison between bacteriophages and human phages, but this part needs some more clarification on the reasons for the comparison. For example, the author says that prophages reactivate in a similar manner to herpes simplex via UV radiation, but does not explain what manner that is.
Minor corrections:
- Line 32, I suggested changing “guessed” to “found” or “discovered”.
- Line 40, change to “they did nothing or even led to the death of the patient because of poor purification, lack of activity”
- Line 41, correct to “the fact that phage therapy”
- Line 49, remove exclamation point.
- Line 68, correct to “After a 30 years abandon in Western”
- Line 70, correct to “Because they were”
- Line 97, correct to “Alive biological transplants”
- Line 112, correct to “Sometimes”
- Line 127, correct to “Even if”
- Line 138, correct to “FMT”
- Line 141, correct to “They certainly skip plasmids, viruses”
- Line 146, please revise sentence.
- Line 151, correct to “Dairy products are a reservoir”
- Line 183, correct to “Various studies have shown that antibiotics”
Author Response
Dear reviewer n°3,
Thanks a lot for your time and comments. I followed your recommandations and corrected all the identified mistakes.
I toned down some comments to minimize harsh statements and documented others with more detailed examples and references, especially on the vaccine side.
The comparison between phages and herpes virus has been more detailed. I also provided more explanations about prophages.
Upon request from another reviewer, you may also see that the introduction has been shorten and focused about the paper in order to get quicker into the matter.
I appreciate the time you spent to review the draft of this paper and want again to warmly thank you.
Kind regards,
Jérôme
Reviewer 3 Report
The present opinion by Jérôme Gabard deals with a very interesting topic and can effectively stimulate discussion in the community. The manuscript is very well written and the author's opinion clearly expressed. I would have recommended publication as is, but for a couple of too harsh statements expressed (especially) about vaccines. Therefore, I'd rather suggest to accept the manuscript after the remarks listed below are fully addressed:
p.4, lines 160-164: "in the case of diet pills….to billions". Please provide references to both the regulations and statistical reports to support the statement.
p.5, lines 193-195: "When it comes…authorization". Please provide a reference to the related regulations.
p.2, line 88. The statement is a bit misleading. Vaccines do not generally bear adventitious contaminants. Nonetheless it's an issue, and it is clearly worth mentioning. Cell substrates that are used to produce vaccines may contain virus-related contaminants. Please change the sentence to a milder expression.
p.5, lines 217-218: "One hundred….by millions". Again, this seems harsh. Please, support your statement with literature evidence, or reword it.
p.5, line 219-222, "However…humans". Please remove this paragraph from the text or provide several references that PCV - or other "pathological retroviruses" - is usually left inside vaccines (please specify the vaccines - more than one? - involved) and reword the sentence to report the scientific evidence on the matter. Ref. 11 is not enough to support such a general statement.
p. 5, line 232-225. "Antibiotics catalyze pro and lysogenic phage release…". Again: please give references in support of the sentence.
Minor points:
p.2, lines 91-92, please add related references (for example: J Virol 2010, doi:10.1128/JVI.02690-09).
p.2, line 99: Merck
p.3, lines 97-100: "As regard… himself". Statement unclear, please rewrite it. Line 103(and 122): sometimes
p.3, lines 107-108: please explain abbreviations (EMA, CMV) when first mentioned in the text.
p.4, line 183 "Various studies have shown"
p.5, line 205 "when the bacterial strain comes straight"
p.5, line 217: "One hundred (of?)and forty years" "have led".
p.5 line 205, "the bacterial strain comes"
Author Response
Dear reviewer n°3,
Thanks a lot for your time and comments. I followed your recommandations and corrected all the identified mistakes.
I provided, as well, complementary references to sustain my thoughts. The number of references is now 42.
I rephrased some paragraphs to minimize harsh statements and documented others with more detailed examples and references, especially on the vaccine side.
Upon request from another reviewer, you may also see that the introduction has been shorten and focused about the paper in order to get quicker into the matter.
I appreciate the time you spent to review the draft of this paper and want again to warmly thank you.
Kind regards,
Jérôme
Round 2
Reviewer 1 Report
This revision is considerably easier to follow than the first draft. English can be improved, but the points are at least understandable.
I would (still) make some organizational changes.
1) First, the thesis of the paper should be stated clearly and early in the paper, set off as the author's main point. My interpretation of the thesis is:
Regulatory agencies are hampering the approval of phage therapy by insisting on the absence of prophage DNA (packaged prophages?) from phage preparations used to treat patients. This insistence sets a double standard, because many other approved biological therapies are allowed far greater levels of viral contamination with no greater evidence of safety.
The first few paragraphs of the Discussion lays out this perspective pretty clearly and could accompany the formal statement of the thesis. This should all go early in the paper so the reader understands what the paper is about. A reason for setting off a formal thesis (e.g., in a box) is both to make the point clear but also to provide something for the Media to quote.
2) I would also expect to see an early, clear statement of the supposed justification regulatory agencies are using to insist on an absence of prophages. Is there no more justification than prophages constituting a contamination -- no reason based on harm? If so, that should be emphasized. I would have thought that the concern is the possible infection of the pathogen by prophages carrying virulence factors (that's a guess). The point of formally adding this to the paper is to help give the reader a perspective on the issues at hand. It will not detract from the Opinion, just help provide a foundation.
A few small points:
3) Many of the claims about the history and the causes (Introduction) are made without reference. While many of us are tempted to interpret history to support our various theses, it is more scholarly to cite someone for major points of historical interpretation.
4) Table 1. Some of the text is white on yellow highlight and I cannot read it. If I truly cared, I'd cut and paste it into Word and change the font to a color I could see.
5) 'Prokaryote bacteria' is not considered a proper taxonomic designation by many. You have Bacteria and Archaea. Some sticklers object to 'prokaryotes.' However, I doubt anyone will care.
6) Line 59: 'sometime essential to their survival' is not in evidence, at least not as a justification for why prophages are found in bacterial genomes. They can be found in bacterial genomes because they got there selfishly.
Note that I am not hiding my identity from the author, but I don't want my name associated with open review. I consider it of little value except when reviews have some important disagreement (despite approving publication) or suggest some major improvement in the paper. Neither is true here.
Author Response
Dear Reviewer,
Thanks for this second round of corrections.
First, I made some organizational changes, including the thesis of the paper stated early and more explanations about the prophage impact.
I also referenced better my historical claims, changed the table text in white to make it black and removed the background blue color.
I must admit that I left the term Prokaryote bacteria although not a proper taxonomic designation by some experts.
I replaced the meaning of “sometime essential to their survival” by saying that prophages may bring genes that support a metabolization process that bacteria need to grow.
I rephrased, as well, some sentences where English was” walking on one leg” instead of two
At last I checked the references numbering and pertinence.
I hope that things are better and acceptable after this second review round .
Thanks again, best regards,
Jérôme
